# The Impact of Childhood and Adult Educational Attainment and Economic Status on Later Depressive Symptoms and Its Intergenerational Effect

**DOI:** 10.3390/ijerph17238970

**Published:** 2020-12-02

**Authors:** Aely Park

**Affiliations:** Department of Social Welfare, Sunchon National University, 255 Jungang-ro, Suncheon, Jeonnam 57922, Korea; apark@scnu.ac.kr

**Keywords:** intergenerational transmission, health inequality, maternal depression, a life-course perspective

## Abstract

This study aimed to investigate a process accounting for the socioeconomic inequality in depressive symptoms from generation to generation. To examine the process, this study utilized data from three generations of grandparents, mothers, and daughters. This study employed data from the Korean Longitudinal Survey of Women and Families, collected from a large-scale national representative sample in South Korea. Conducting pathway analysis, the study tested direct and indirect pathways between mother’s socioeconomic status (SES) experienced in childhood and their offspring’s depressive symptoms through maternal SES and depressive symptoms in adulthood. This study found that early economic hardship increased the risk of depressive symptoms in daughters through maternal low education and depressive symptoms (*β* = 0.03, *p* < 0.05), which was consistent with the theoretical framework, which relied on a life-course model highlighting that early life experiences affect later adult health and can potentially have effects across generations. This finding suggests that interventions that work with maternal education and depression may benefit from efforts to break the likelihood of continuity of depressive symptoms into the next generation, especially for their own daughters.

## 1. Introduction

It is well documented that SES creates an environment that is essential for the preservation of health, and it is known as an important determinant of physical health as well as psychosocial stress such as depressive symptoms [1,2,3,4,5,6]. Previous studies confirmed that people with low income and low educational attainment are more likely to experience health problems than those with high SES, and the evidence has been supported across a range of health measures and different social groups [3,7,8]. The difference in health status according to SES has been discussed as the concept of health inequality. That is, health inequality refers to differences in health that is closely linked with socioeconomic disadvantage, and it adversely affects individuals, or groups who have systematically experienced worse health based on their income, education attainment, or both [9,10].

Discussion on the difference in health level according to SES has been mainly dealt with in advanced countries in physical health in the 1980, and this phenomenon has been conceptualized as health inequality [11]. In particular, educational attainment, occupation, and income levels are known to be important factors for physical health inequality [1,3,5]. In South Korea, it has been reported that approximately 30% of the direct factors of physical health inequality resulted from income disparities [6]. Recently, some studies have pointed out that mental health problems might be linked to health inequality in several respects. First of all, mental health inequality possibly begins in childhood. For example, children of families suffering from poverty showed more psychological and emotional difficulties such as depressive symptoms and anxiety than children of non-poverty families [12,13]. Further, a few important studies revealed that adverse SES in childhood significantly connected to negative mental health in adulthood later [8,14,15]. In addition, women are more susceptible to depressive symptoms. For example, the number for depressive symptoms for women showed doubling of odds than that for men having depressive symptoms and, women with low socioeconomic status experienced more stressful events and a quarter of women with low socioeconomic status suffered from depressive symptoms in South Korea [16,17]. Based on the information, it can be presumed that mental health problems might be connected to adverse childhood experiences and substantially more common among women in low socioeconomic status.

However, the previous studies on the relationships between childhood SES and health mainly focused on physical health, and there are relatively few studies on mental health. Thus, it is unclear whether such relationships exit in mental health [8]. To fill the gaps in current knowledge, this study examines whether childhood SES is related to depressive symptoms later using data from the Korean Longitudinal Survey of Women and Families. This study considers a life-course perspective as a tool for understanding how early-life adversity shapes a later mental health outcome.

### 1.1. Life-Course Perspective

A life-course perspective focuses on understanding of how early-life experiences affect later adult health and potentially affecting across generations [9,18]. This approach is relevant to understanding and dealing with health inequalities, because it focuses on the role of contextual factors in relation to socioeconomic inequalities in health [18]. The life-course perspective reflects on both prior and current conditions affecting health status, and this approach also indicated intergenerational effects [9,18].

This approach recognizes that differences in early life exposure have a profound impact on later health regardless of subsequent life events. It implies a crucial time during the life course that critically or permanently impacts on later health and it generally emphasizes on childhood as a window of time during which health is affected in ways that could have unalterable changes, e.g., very poor nutrition in childhood. In addition, this approach focuses on the health effects resulting from damages generate through a long-term exposure to adverse conditions over the life course. Even those as distant as parental SES may be important predictors to explain current health outcomes throughout lagged exposures. In relation to the notion, when social mobility is low, prior life conditions may be powerful predictors of explaining current health outcomes of individuals who grew up poor. Finally, intergenerational effects refer to the transmission of conditions and its outcomes such as health or ill health across more than one generation. Thus, health is viewed as a consequence of intergenerational transmission of social status [9,18,19,20]. Important, but few, studies on the life course perspective have been published. This study provides relevant studies investigating the effects of SES on depressive symptoms addressing health disparities.

### 1.2. The Effects of Childhood SES on Depressive Symptoms Later

Mental health is an issue emerging as an important social problem, and depressive symptoms are one of the more prevalent mental health problems in South Korea [17]. Mental health problems prevalent in Korean women may deprive them of the pleasure of life and lower their motivation for their entire life [21]. Previous studies assumed that stressful experiences related to economic adversity and the social disadvantages may contribute to poor mental health including depressive symptoms [22,23]. Growing research found variations in depressive symptoms by SES [3,4,22,23,24,25,26,27,28]. In a meta-analysis, the study found moderate to strong links between SES and depressive symptoms, and low SES has been associated with more severe and persistent depressive symptoms in their analysis [3,27].

Life-course studies revealed that both childhood economic adversity and low parental education are associated with an increased risk of depressive symptoms for middle-aged women [23,29]. In addition, their own educational attainment and family income level in adulthood significantly linked to depressive symptoms later [8,15]. One study found that women in childhood adversity and economic disadvantage had about a doubling of odds for experiencing later depression [30]. However, the effect of adult education is stronger in middle-aged people than on the effect at a family income level [31,32]. Empirical studies on the relative contributions of earlier and later SES conditions to depressive symptoms are rare, and the findings are not consistent. For example, some empirical studies presented supporting evidence that adverse SES in childhood significantly connected to depressive symptoms in adulthood once adult SES was taken into account [8,33]. Some studies did not find the lasting effects of adverse SES in childhood on depressive symptoms in adulthood controlling for adult SES [34]. Few, but meaningful, studies found that adverse SES in childhood is significantly and negatively related to both their SES conditions in adulthood and later depressive symptoms, and also, adverse SES in childhood is significantly related to later depressive symptoms throughout their SES conditions in adulthood [14,15,33].

### 1.3. Intergenerational Transmission of Adverse SES and Depressive Symptoms

A life-course perspective also highlights that early life experiences affect later adult health and can potentially affect across generations [18]. It is well known that maternal depressive symptoms are associated with an increased risk of daughter’s depressive symptoms [35,36,37,38,39]. One study discussed a mechanism where the intergenerational transmission of depression from mothers to daughters was the long-term exposure to stressors, which may be set in motion in the early years of life and, in turn, predict offspring depressive symptoms [37]. In addition to the mechanism, one study reviewed that intergenerational transmission of depressive symptoms is more typical to female behavior than male throughout the developmental process [10]. As indicated in the previous literature, females tend to be more relationship-orientated and more reactive to interpersonal stress generally and, thus, these risk patterns suggest that females’ interpersonal orientation may be related to inner directed distress such as depressive symptoms [40,41].

There are only a few studies indicating the intergenerational effects of adverse SES on depressive symptoms. For example, one study found that low parental education predicted offspring’s adult depressive symptoms after parental depressive symptoms were accounted for [42]. In addition, in South Korea, one study examined the effects of low parental education on offspring’s adult SES and their mental health problems [15]. They found that low parental education affected a lower educational level of offspring, which contributed to later depressive symptoms.

However, though a robust body of literature suggests depressive symptoms in mothers may lead to daughters’ depressive symptoms, researchers have paid less attention to how SES differences in depressive symptoms is transmitted from one generation to next generation.

### 1.4. Purpose

Since the relationships still remain unclear, it is examined whether SES in childhood is related to later depressive symptoms, and also, to try to examine whether such effects are prolonged into the next generation throughout adult SES and depressive symptoms in previous generations. This study considers a process accounting for the transmission of socioeconomic problems from generation to generation through SES and depressive symptoms. This study considers the following research questions: First, does childhood SES have lasting effects on depressive symptoms in later life controlling their adult SES? Second, does adult SES mediate the relationship between childhood SES and depressive symptoms later? Third, does childhood SES have intergenerational effects on depressive symptoms in the next generation throughout one’s own depressive symptoms?

## 2. Method

### 2.1. Data and Sample

This study employed data from the Korean Longitudinal Survey of Women and Families (KLoWF) collected by Korean Women’s Development Institute in South Korea. The KLoWF is a longitudinal survey of nationally representative Korean women, and 9997 women aged from 19 to 64 who lived in 9068 households across the nation were interviewed for the first time in 2007. The first wave survey (2007) and second wave survey (2008) were conducted every year. Following surveys were made every other year in 2010 and 2012.

This study considers wave 1 as the baseline wave and analyze data from the first and fourth waves (2007, 2012) of KLoWF, which contains measures for the variables of interest in this study. For example, KLoWF collected data for depressive symptoms for mothers and daughters at wave 4. To test intergenerational transmission of health inequality, this study restricted the sample of pairs of mothers and daughters who participated at both waves 1 and 4. In addition, to test research questions, this study used available data collected from KLoWF, regarding information on the grandparents’ education attainment and economic status (Generation1, G1), mothers’ education attainment and their depressive symptoms (G2), and daughters’ depressive symptoms (G3). The G1 information came from mothers’ data. The case numbers of both mothers and their daughters participating at wave 1 was 651, and the case numbers of both mothers and their daughters participating at wave 4 was 189. This study restricted the sample of pairs of mothers and their daughters who participated at both waves 1 and 4. Thus, the resulting sample included 189 mothers and 189 daughters participating at waves 1 and 4, who accounted for 3.8% of the 9997 cases.

### 2.2. Measures

*Socioeconomic status (G1):* The mothers (G2) reported G1′s socioeconomic status about the time when they were around 15 years of age in wave 1. The grandfather’s education (G1) was divided into two groups: less than primary school graduation (= 1) and primary school graduation or above (= 0). Grandmother’s education (G1) was measured in the same manner as done for the grandfather’s education. The mothers also reported their grandparent’s economic status in wave 1 and it was measured with one question “what was the economic circumstance of your family about the time when you were around 15 years of age?” (1 = very affordable, 5 = very poor).

*Mothers’ outcomes (G2).* The mothers’ year of education measured in wave 1 was a continuous variable. Their education attainment is the highest level of education attained by G2 respondents. Mothers reported their economic status in wave 1 and it was measured with one question “what is the current economic circumstance of your family?” (1 = very affordable to 5 = very poor). In addition, the mother’s depressive symptoms was measured in wave 4. KLoWF measured depressive symptoms using the Center for Epidemiologic Studies Depression Scale (CES-D) as adapted for use with Korean adults [43,44]. The CES-D uses 10 four-point Likert scale questions (1 = not at all true, 4 = all the time) regarding cognitive, affective, behavior, and somatic symptoms of depression experienced during the past week (Cronbach’s alpha = 0.81). Higher scores on the scale indicated greater levels of depressive symptoms.

*Daughters’ outcome (G3).* The daughter’s depressive symptoms were measured in wave 4. Their depressive symptoms were measured in the same manner as was done for the mothers. KLoWF measured depressive symptoms using the Center for Epidemiologic Studies Depression Scale (CES-D) as adapted for use with Korean adults [43,44]. The CES-D uses 10 four-point Likert scale questions (1 = not at all true, 4 = all the time) about cognitive, affective, behavior, and somatic symptoms of depression experienced during the past week (Cronbach’s alpha = 0.78). Higher scores on the scale indicated greater levels of depressive symptoms.

*Control variables:* This study employed control variables measured in wave 1 to account for socioeconomic and heath inequality of the study respondents. The mother’s age was a continuous variable measured in years. This study coded employment status as “currently employed” (= 1) for those who were employed and paid for their work and “not employed” (= 0) for those who were unemployed and were not looking for a job. The mother’s living arrangement with a spouse was a dichotomous variable indicating whether they were living with their spouse (= 1) or not (= 0). The mother’s number of siblings was a continuous variable measured in counting the number of siblings who grew up together. The mother’s household yearly income was calculated as the total value of income (in thousands, Korean Won) earned by all household members for the past calendar year. The daughter’s age was a continuous variable measured in years, and their education was a continuous variable of the highest level of education attained by the daughter (G3) respondents. The number of siblings of the daughters was a continuous variable measured in counting siblings who grew up together. The daughter’s employment status was measured in the same manner as was done for the mothers.

### 2.3. Analysis

To examine the research questions, analyses were conducted as follows. First, this study conducted descriptive statistics to summaries sample characteristics and check non-normality in variables by examining univariate skewness and kurtosis. Second, to examine the research questions, this study conducted bivariate analysis to check whether socio-economic factors were associated with depressive symptoms without considering other covariates. This study then employed path model analysis to explore intergenerational transmission of health disparity between mothers and daughters using Mplus [45]. Path analysis is an appropriate method to model mediation, as it solves an entire set of regression equations simultaneously and enables calculation of direct and indirect effects between measures. Path model analysis provides a number of indices that can be used to determine the goodness of fit of the model to the data. Overall goodness of fit for the models was assessed using the χ^2^ likelihood ratio statistic, root mean square error of approximation (RMSEA), Bentler’s normed comparative fit index (CFI), the Tucker Lewis Index (TLI) [29], and component model fit, which is assessed using modification indices. The significance level (α) is the probability of the study rejecting the null hypothesis when the null hypothesis is true [46], and 0.01, 0.05, and 0.01 are the most commonly used values for the significance level, representing a 1%, 5%, and 10% chance of type I error occurring [47]. Thus, our research design allowed us to identify the effects that the socioeconomic status experienced by the grandparents had on the depressive symptoms of the daughters through their mothers’ depressive symptoms.

## 3. Results

### 3.1. Descriptive Results and Bivariate Tests

Table 1 showed the descriptive statistics for selected and unselected samples. In addition, it showed *t*-test and chi-square test results to see if there is a group difference between selected and unselected samples, and there were no group differences in the variables used. In terms of descriptive statistics for selected sample, the mean economic status was 3.2 among 5, indicating that the majority of G2 mothers considering the economic status of their family about the time when they were around 15 years of age was normal. The mean year of schooling for G2 mothers was approximately 11, and about 45% of mothers did not complete high school or its equivalent. The mean of the G2 mother’s depressive symptoms was 9.7 (SD = 4.6) in wave 4, and the mean of the G3 daughter’s depressive symptoms was 8.6 (SD = 4.5) in wave 4, indicating that the average levels of depressive symptoms of both mothers and their daughters were not high. The average age of G2 mothers was about 50 years old and the mean number of G2 mother’s sibling was about 5. The average amount of household income of G2 mothers was 1907 (10,000) KRW per year, roughly equivalent to USD 17,026. In addition, the average age of G3 daughters was about 24 years old, and the mean number of G3 daughter’s sibling was about 2.4. The mean year of schooling for G3 daughters was about 15, and only one out of 189 G3 participants did not complete high school. In addition, G2 mothers reported that about 19% of grandfathers had not graduated from primary school and about 31% of grandmothers had not graduated from primary school. In addition, about 48% of G2 mothers were not currently employed, and almost 80% of G2 mothers were living with their spouse. Finally, approximately 43% of daughters were currently employed.

Bivariate associations between SES and depressive symptoms controlling for covariates were shown to determine whether the level of education and economic status of one’s own or previous generation are related to depressive symptoms of one’s own or later generation in Table 2. While less than primary school graduation of both G1 grandfather and grandmother were not related to depressive symptoms across generations, a significant relationship existed between G2 mother’s year of education and their depressive symptoms (*β* = −0.27, *p* < 0.001). However, G2 mother’s year of education was not related to G3 daughter’s depressive symptoms. In addition, significant relationships existed between G1 grandparent’s economic status and the subsequent generations of depressive symptoms. G1 grandparent’s economic status was significantly and positively related to G2 mother’s depressive symptoms (*β* = 0.22, *p* < 0.01), and G3 daughter’s depressive symptoms as well (*β* = 0.18, *p* < 0.001). However, G2 mother’s economic status was not related to their depressive symptoms and their daughter’s depressive symptoms.

### 3.2. Testing the Mediating Effects

Next, path analyses were conducted to examine the associations among intergenerational factors and outcomes. To test research questions 1 and 2, Figure 1 shows a path model by mother’s years of education (G2) and economic status (G2) as mediators. It presents standardized estimates for the pathways from G1 socioeconomic status to G2 depressive symptoms, controlling for covariates. The resulting model showed a reasonable fit to the data (χ^2^ (11) = 59.98, CFI = 0.91, RMSEA = 0.06, N = 189). The significant pathways are as follow: Both G1 grandfather’s and grandmother’s primary school graduation was significantly related to G2 mother’s economic status (*β* = 0.19, *p* < 0.05, *β* = 0.26, *p* < 0.01, respectively), indicating that G2 mothers who grow up with parents who had not graduated primary school recognized that their current economic status was worse than others. G1 economic status significantly affected G2 mothers’ year of education (*β* = −0.30, *p* < 0.001), but was significantly related to both G2 mother’s economic status (*β* = 0.16, *p* < 0.05) and G2 mothers’ depressive symptoms (*β* = 0.14, *p* < 0.05), indicating that the more G2 mothers perceived being poor at the time when they were 15 years of age, the less educational attainment, the less affordable, but higher depressive symptoms they experienced in middle age. In addition, G2 mothers’ year of education significantly impacted on their depressive symptoms (*β* = −0.20, *p* < 0.05), indicating that higher educational attainment was related to lower levels of depressive symptoms in middle age.

Then, to test research question 3, Figure 2 shows a path model that employed G1, G2, and G3 variables. It presents standardized estimates for the pathways to test whether childhood SES has intergenerational effects on depressive symptoms in the next generation throughout one’s own depressive symptoms. The resulting model showed a reasonable fit to the data (χ^2^ (9) = 12.54, CFI = 0.95, RMSEA = 0.04, N = 189). G1 economic status was significantly related to G2 mothers’ depressive symptoms (*β* = 0.21, *p* < 0.01), indicating that the more G2 mothers perceived being poor about the time when they were 15 years of age, the higher depressive symptoms they experienced in middle age. In addition, G2 mother’s depressive symptoms was significantly related to G3 daughter’s depressive symptoms (*β* = 0.45, *p* < 0.001), indicating that mother’s depressive symptoms are strongly related to their daughter’s depressive symptoms in next generation. Thus, it revealed intergenerational continuity of depressive symptoms between G2 mothers and G3 daughters.

Finally, this study added the Figure 2 model into the Figure 1 model to test whether G1 education and economic status increases the risk of cascading consequences that lead from the response of the mother’s generation to being entrenched in the next daughter generation. Figure 3 shows a final path model that employed G1, G2, and G3 variables. It presents standardized estimates for the pathways from G1 educations and socioeconomic status to G3 depressive symptoms, controlling for covariates. The resulting model showed a reasonable fit to the data (χ^2^ (18) = 33.01, CFI = 0.92, RMSEA = 0.06, N = 189). The patterns of significant relationship between intergenerational variables were the same as in the previous path models. This study found that low economic status in childhood increased the risk of low educational attainment in adulthood and higher depressive symptoms, which in turn increase the risk of depressive symptoms in their daughters.

Mediation can be confirmed by examining whether one variable influences another variable through a third variable. To assess the presence of mediation, a joint significant test was performed in this study [41,48]. In other words, the estimated parameters for each portion of the indirect path are multiplied (path a × path b), and it demonstrated a significant indirect effect a × b. A direct effect is that the original predictor is related to final outcome, controlling for the mediating variable.

Table 3 showed the results of indirect effects in the pathways. The first pathway from G1 socioeconomic status to G2 depressive symptoms showed that G1 economic status had a significant positive indirect effect on G2 depressive symptoms through G2 year of education (*β* = 0.07, *p* < 0.05). Thus, G1 grandparents’ economic status was associated with G2 mothers’ year of education, which in turn increased the risk of G2 mothers’ depressive symptoms. This study also examined the pathway from G1 socioeconomic status to G3 depressive symptoms. The results showed that the indirect effect of G2 mothers’ depressive symptoms on the association between G1 grandparents’ economic status and G3 daughters’ depressive symptoms was significant (*β* = 0.06, *p* < 0.05). Thus, G1 grandparents’ economic status increased the risk of G2 mothers’ depressive symptoms, which in turn increased the risk of G3 daughters’ depressive symptoms. Finally, another indirect effect was found, as the G1 grandparents’ economic status was associated with G2 mothers’ year of education, which increased the risk of G2 mothers’ depressive symptoms, and in turn increased the risk of G3 daughters’ depressive symptoms (*β* = 0.03, *p* < 0.05). 

## 4. Discussion

The current study sought to advance the understanding how adverse SES in childhood links to later mental health, depressive symptoms, and how such effects reach into the next generation. Very few studies have investigated the intergenerational effects of SES on depressive symptoms across generations. This study considered a process accounting for the transmission of socioeconomic problems from generation to generation through SES and depressive symptoms. To examine the process, this study utilized the data from mothers and their daughters. This study first examined whether the mother’s early life SES experienced in childhood has direct effects on the mother’s later depressive symptoms and have if it has lasting effects on their daughter’s depressive symptoms. Then, this study examined whether adverse childhood SES increases the risk of cascading consequences that lead from a response of the mother generation to being entrenched in the next daughter generation. The pathway still remains unclear, and there is little empirical evidence outside of western developed countries regarding these complicated relationships. Thus, this study conducted to fill the current knowledge gap by using longitudinal data collected in South Korea.

This study found that childhood economic status and adult economic status had independent effects on the mother’s depressive symptoms. Lower childhood economic status was associated with higher depressive symptoms in later life. The effect of childhood economic status on adult depressive symptoms operated through its effect on educational attainment in adulthood. Our findings support previous research, suggesting that exposure to economic hardship in early life predicted higher levels of depressive symptoms later [2,4,15,16,25]. In addition, our findings support previous research that childhood SES determines the living conditions in adult life, which gives rise to health inequality [8,34]. In addition to the information, a systematic review of the epidemiological study of the last 19 years supports that the social and economic conditions of poverty such as low income and less education is a consistent positive association with mental health problems [29], and this pattern could also be found through community-based studies of other Asian countries such as Japan, India, Pakistan, Taiwan, and Turkey (listed alphabetically) [4,49,50,51,52].

In addition, our study added a significant finding that the effect of mother’s economic hardship persisted into the daughter generation. The findings are consistent with the theoretical frameworks relying on a life-course model, which highlights that early life experiences affect later adult health and potentially across generations [5,18]. The evidence suggests that economic inequality on mental health, at least depressive symptoms, can be reproduced in successive generations, especially for women. In South Korea, economic hardship broadly shapes exposure to chronic stressors that lead to mental health problems across several lifespans [2,7]. In addition, income transmission across generations is generally large in low income groups and being poor implies a much lower rate of transmission of economic advantage to the next generation [2,16]. However, studies such as the one that follows economic inequality on mental health across generations are rare, and more studies need to replicate whether the intergenerational pattern is typical for women. It is difficult to make a firm conclusion about the health inequality in our study.

Evidence from the meditational models supports the mechanism underlying the association between early economic hardship and later depressive symptoms. This study found that the mother’s economic hardship in childhood increased the risk of depressive symptoms in daughters throughout maternal low educational attainment and depressive symptoms. First of all, the pathway proposes that maternal depressive symptoms may be salient in accounting for links between the history of economic hardship and depressive symptoms among daughters. This finding is consistent with previous literature, which suggests that offspring of depressed mothers are more likely to experience mental health problems [40,41]. In addition, it is noteworthy that the mechanism underlying the association can be delineated in the pathway through maternal education attainment. However, maternal education attainment was not directly related to the daughters’ depressive symptoms but was related to maternal depressive symptoms. The result showed that the family’s financial well-being in childhood made a difference in the educational attainment rather than the perception of economic well-being in the mother’s generation. Along with others, this study found some indication that a poor environment limits the opportunity to receive education that is a protective factor for mental health [15,53]. In addition, one study discussed that education is a significant factor promoting coping resources and social support, which may protect depressive symptoms [53]. Thus, our study indicated that early economic hardship significantly contributes to later depressive symptoms and that education levels play a role in mediating this relationship in a long-term perspective. Although this study did not compare the pathways between males and females in this study due to the limited nature of the data, women generally had fewer opportunities for upward mobility, and they also had poorer mental health in South Korea [15,17]. That is, even if the childhood environment is unfavorable, there may be less room for women to increase educational achievement and mitigating adverse effects of childhood environment on depressive symptoms [8]. Thus, it suggests that benefits of upward mobility through better education for women may reduce inequality in SES and depressive symptoms.

To further address the potential knowledge of mental health inequality, more research is warranted to identify the pathways from early life adversity to mental health problems that pass from parents to their offspring generations. In addition, evidence from the meditational models suggests a potential point of intervention intended to break the intergenerational cycle. Mothers with history of adverse economic status have an increased risk of depressive symptoms, which may reach into their middle age and pose a risk for a further depression cycle with the next generation. Thus, preventive interventions that work with maternal depression may benefit from efforts to break the likelihood of continuity of depressive symptoms into next generation, especially for their own daughters. In addition, it is important that when working with maternal depression, one needs to clearly understand the social status context, which could potentially be harmful to maternal and their offspring’s mental health. Providing more opportunities for upward mobility through better education might be one of the ways to reduce inequality in mental health. Taken together, the current study extends the current knowledge to address inequality in mental health underlying in the continuity of depressive symptoms across generations.

This study is not free from limitations. The KLoWF was not designed to investigate the intergenerational study, and, thus, this study restricted the sample of pairs of mothers and daughters who participated in surveys, which resulted in a small number of cases. It would be desirable to replicate the study with larger samples. In addition, G1 socioeconomic variables were measured in a retrospective manner, where G2 mothers reported G1′s socioeconomic status about the time when they were 15 years of age, which may have introduced biases such as recall bias. In addition, because the KLoWF collected data around G2 mothers, the information on G3 is limited. Since the KLoWF did not collect the information on the depressive symptoms from G1 grandfathers and grandmothers, this study could not confirm whether depressive symptoms for mothers and daughters derived from G1 generation. The study may also have omitted some important variables of socioeconomic position, particularly occupation and wealth. This would allow evaluating depressive symptoms through socioeconomic position in relation to these additional measures. In addition to the limitations, the KLoWF did not provide clinical information about general and mental health around mothers and daughters.

## 5. Conclusions

Despite these limitations, this article contributes to literature on the inequality in mental health and intergenerational transmission of depressive symptoms. To our knowledge, this is one of the first studies to use a longitudinal sample of mothers and daughters to examine the effect of early life adversity on mental health across generations in South Korea. The current study suggests that the study of mental health inequality needs to be expanded to include intergenerational continuity of stress and depressive symptoms, especially for women over long periods of time.

## Figures and Tables

**Figure 1 ijerph-17-08970-f001:**
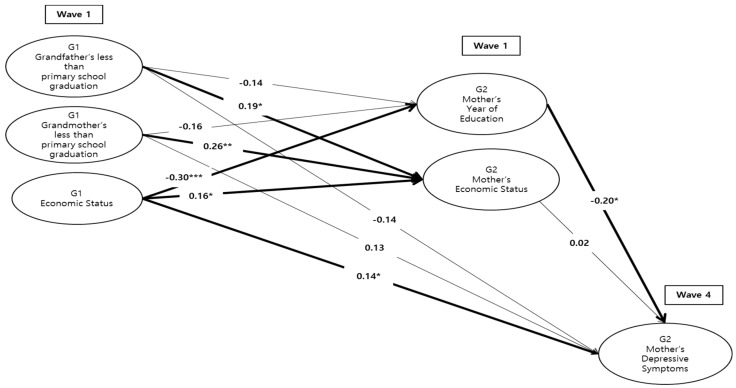
Standardized results of intergenerational model by mother’s years of education (G2) and economic status (G2) as mediators. The model above controls mother’s demographic factors such as mother’s age, mother’s living with spouse, mother’s number of siblings, mother’s family income, and mother’s employment. The thick line in the figure indicates statistical significance. * *p* < 0.05, ** *p* < 0.01, *** *p* < 0.001.

**Figure 2 ijerph-17-08970-f002:**
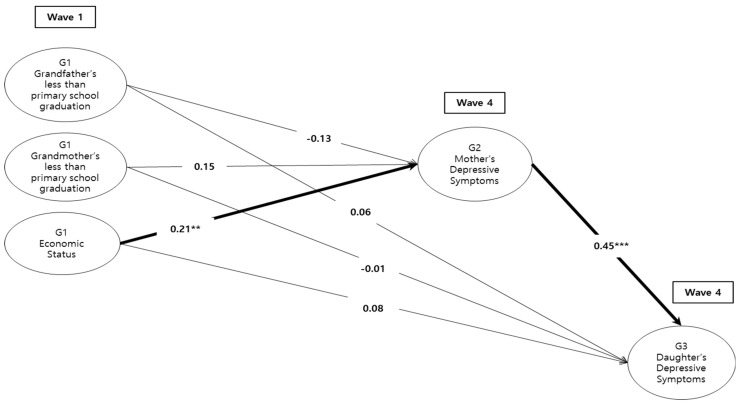
Standardized results of the intergenerational model by the mother’s depressive symptoms (G2) as a mediator. The model above controls mother’s and daughter’s demographic factors such as mother’s age, mother’s living with spouse, mother’s number of siblings, mother’s family income, and mother’s employment, daughter’s age, daughter’s year of education, daughter’s number of siblings, and daughter’s employment. The thick line in the figure indicates statistical significance. ** *p* < 0.01, *** *p* < 0.001.

**Figure 3 ijerph-17-08970-f003:**
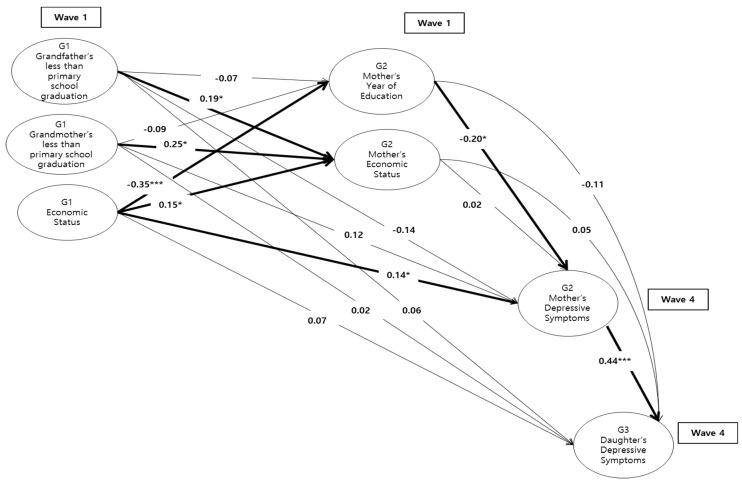
Standardized results of intergenerational model by mother’s depressive symptoms (G2), Table 2 and economic status (G2) as mediators. The model above controls mother’s (G2) and daughter’s (G3) demographic factors such as mother’s age, mother’s living with spouse, mother’s family income, mother’s number of siblings, mother’s employment, daughter’s age, daughter’s year of education, daughter’s number of siblings, and daughter’s employment. The thick line in the figure indicates statistical significance. * *p* < 0.05, *** *p* < 0.001.

**Table 1 ijerph-17-08970-t001:** Descriptive statistics for selected and unselected samples.

Variables	Selected (*n* = 189)	Unselected (*n* = 462)	*t*-Value	*p*-Value
	M(SD)	M(SD)		
G1 Economic Status (range = 1–5)	3.2 (0.9)	3.3 (0.9)	1.86	*p* > 0.05
G2 Mother’s year of education (range = 1–18)	10.6 (2.9)	10.0 (3.1)	−1.38	*p* > 0.05
G2 Mother’s depressive symptoms (range = 0–24)	9.7 (4.6)	9.9 (4.8)	0.55	*p* > 0.05
G2 Mother’s age (range = 40–63)	50.2 (5.1)	50.8 (5.3)	1.27	*p* > 0.05
G2 Mother’s number of sibling (range = 1–10)	5.4 (1.7)	5.2 (1.8)	−1.20	*p* > 0.05
G2 Mother’s household yearly income	1906.9 (1207.1)	1893.1 (1239.7)	0.13	*p* > 0.05
(W, ten-thousands) (range 100–8000)				
G3 Daughter’s depressive symptoms (range = 0–20)	8.6 (4.5)	9.5 (3.8)	1.50	*p* > 0.05
G3 Daughter’s age (range = 19–39)	23.8 (4.3)	24.6 (4.2)	1.94	*p* > 0.05
G3 Daughter’s number of sibling (range = 1–5)	2.4 (0.7)	2.5 (0.9)	1.72	*p* > 0.05
G3 Daughter’s year of education (range = 9–22)	14.6 (1.8)	14.4 (1.9)	−0.89	*p* > 0.05
	(%)	(%)	x^2^	
G1 Grandfather’s less than primary school graduation	19.0	24.1	1.85	*p* > 0.05
G1 Grandmother’s less than primary school graduation	31.2	34.8	0.75	*p* > 0.05
G2 Mother’s employment (employed)	48.1	43.1	1.40	*p* > 0.05
G2 Mother’s living with spouse (yes)	78.8	84.4	2.93	*p* > 0.05
G3 Daughter’s employment (employed)	43.4	49.6	2.05	*p* > 0.05

Note. M = Mean, SD = Standard deviation for continuous variables.

**Table 2 ijerph-17-08970-t002:** Bivariate relationship between socioeconomic factors and depressive symptoms controlling for covariates (each regression coefficient).

Socioeconomic Factors	Depressive Symptoms	
	G2 Mothers	G3 Daughters
G1 grandfather’s less than primary school graduation	0.003	0.04
G1 grandmother’s less than primary school graduation	0.12	0.07
G1 economic status	0.22 **	0.18 ***
G2 mother’s year of education	−0.27 ***	−0.13
G2 mother’s economic status	0.47	0.09

Note. Standard coefficient is shown. The models control for mother’s and daughter’s demographic factors such as mother’s age, mother’s living with spouse, mother’s number of sibling, mother’s family income, mother’s employment, daughter’s age, daughter’s number of sibling, daughter’s year of education and daughter’s employment.** *p* < 0.01, *** *p* < 0.001.

**Table 3 ijerph-17-08970-t003:** Indirect effects for the mediational model.

Pathways	Standardized Estimate	Standard Error	*t*-Value
*From* *G1 SES to G2 Depressive Symptoms*			
G1 economic status→G2 year of education→G2 depression	0.071 *	0.031	2.276
*From* *G1 SES to G3 Depressive Symptoms*			
G1 economic status→G2 depression→G3 depression	0.061 *	0.031	1.990
G1 economic status→G2 year of education→G2 depression→G3 depression	0.031 *	0.015	2.158

Note: Standard coefficients are shown. * *p* < 0.05.

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
