# Peer review of "The Impact of Childhood and Adult Educational Attainment and Economic Status on Later Depressive Symptoms and Its Intergenerational Effect"

_ijerph, 2020, doi:10.3390/ijerph17238970_

Round 1
Reviewer 1 Report
The present manuscript aimed to investigate a process accounted for socioeconomic inequality in depressive symptoms from generation to generation, its main contribution is found that the economic status of childhood and the economic status of adults had independent effects on the mother's depressive symptoms and added a significant finding that the effect of the mother's financial difficulties persisted in the daughter's generation.
The manuscript is generally well written, however some recommendations and questions are the following
The Methods section should include a statement indicating that the research was approved by an independent ethics committee
The methodology is not easy to read, I consider that a diagram that explains which evaluations were made to each group and wave would help to clarify it, for example for G2 the SES was evaluated in wave 1 and the depressive symptoms in wave 4?
Do daughters and mothers live at the same address? Was this a selection criterion?
Why was the annual family income of the daughters not considered ?
Could you explain why the same criteria were not taken to evaluate the SES in G1, G2 and G3?
In the Analysis section please describe the alpha level that defines statistical significance
In table 1 the p-values ​​for each result of the statistical test must be added
In line 313, corresponding to the legend of figure 3 instead of Table 2 it should say "economic status"
Figure 3 should improve its meaning, it is very useful that the thickness of the lines is greater in the statistically significant results, however it seems that some values ​​without statistical significance are presented as bold letters, in any case the meaning of the thickness of the lines should be explained in the figure legend
Author Response
Responses to Reviewer’s Comments
The Impact of Childhood and Adult Educational Attainment and Economic Status on Later Depressive Symptoms and It’s Intergenerational Effect
(Manuscript ID: IJERPH-990995)
I appreciate the reviewer’s constructive feedbacks that held me clarify the important point.
1. The Methods section should include a statement indicating that the research was approved by an independent ethics committee
The KLoWF is an old panel established in 2006, and at that time, the researchers were not recognized the importance of undergoing IRB-review. Thus, it was not approved by the institutional IRB yet. However, it is widely used as a representative survey of Korean women in Korea, and institutional IRB screening is currently underway.
2. The methodology is not easy to read. I consider that a diagram that explains which evaluations were made to each group and wave would help to clarify it, for example for G2 the SES was evaluated in wave 1 and the depressive symptoms in wave 4?
In each figure, information on the wave of the variable is provided.
3. Do daughters and mothers lie at the same address? Was this a selection criterion?
This was not a selection criterion. Although KLoWF did not provide information about the address, if any of the women interviewed in the secondary data had daughters, they were found and interviewed.
4. Why was the annual family income of the daughters not considered? Could you explain why the same criteria were not taken to evaluate the SES in G1, G2 and G3?
KLoWF is collected around G2 mothers and the information on G3 is limited. G1 SES was reported by G2 mothers so that G1 SES was based on G2 mothers experience before age 15. The KLoWF did not provide the information about the annual family income of daughters. I described it as a study limitation in the discussion section.
5. In the Analysis section please describe the alpha level that defines statistical significance
I have added the sentences in the Analysis as the following:
“The significance level (α) is the probability of the study rejecting the null hypothesis when the null hypothesis is true(Dalgaard, 2008), and .01, .05 and .01 are the most commonly used values for the significance level, representing a 1%, 5%, and 10% chance of type I error occurring(Lavrakas, 2008).”
6. In table 1 the p-value for each result of the statistical test must be added
The table 1 showed if there is a group difference between selected and unselected samples, and there was no group difference in the variables used. Thus, p-value in each t-test and chi-square test is over .05 (p > .05).
7. In line 313, corresponding to the legend of figure 3 instead of Table 2 it should say “economic status”
The line 313 is the section describing control variables using in the Figure 3 model.
8. Figure 3 should improve its meaning, it is very useful that the thickness of the lines is greater in the statistically significant results, however it seems that some value without statistical significance are presented as bold letters, in any case the meaning of the thickness of the lines should be explained in the figure legend
It was noted that in each figure, the bold line indicates statistical significance.
Reviewer 2 Report
The article authored by Aely Park aimed at investigating a process accounted for socioeconomic inequality in depressive symptoms from generation to generation. The data from the Korean Longitudinal Survey of Women and Families (KLoWF) collected from a large-scale national representative sample in South Korea were employed for analyses. The Author found that early economic hardship increased the risk of depressive symptoms in daughters through maternal low education and depressive symptoms.
The study and statistical analyses seem to be well done. However, before being accepted, I would suggest the author provide more details regarding the following points:
- The results of the study show the relationship between childhood socioeconomic status and mental health. However, as the KLoWF is representative for Korean women the results are specific and limited to Korean nation. It would be useful to discuss briefly what the situation is in other countries (Asian and / or European) in light of the results for the Korean population.
- In the Discussion section, lines 416-417 , the Author mentioned that ”… the number of cases used in the analysis is relatively small”. Please provide the reader with some numerical data in this context, please tell in how many cases (out of the analyzed cases) the intergenerational effect of socioeconomic status on depression symptoms was found.
Author Response
Responses to Reviewer’s Comments
The Impact of Childhood and Adult Educational Attainment and Economic Status on Later Depressive Symptoms and It’s Intergenerational Effect (Manuscript ID: IJERPH-990995)
I appreciate the reviewer’s constructive feedbacks that held me clarify the important point.
1. The results of the study show the relationship between childhood socioeconomic status and mental health. However, as the KLoWF is representative for Korean women the results are specific and limited to Korean nation. It would be useful to discuss briefly what the situation is in other countries (Asian and / or European) in light of the results for the Korean population.
I think it is very useful to describe the situation in other countries. I put the sentences in the discussion section of this manuscript:
“In addition to the information, a systematic review of the epidemiological study of the last 19 years supports that the social and economic conditions of poverty such as low income and less education consistent positive association with mental health problems (Lund, et al., 2010), and this pattern could also be found through community-based studies of Asia countries such as Japan, India, Pakistan, Taiwan, and Turkey (listed alphabetically) (Gulseren et al., 2006; Maselko, et al., 2018; Nishimura, 2011; Patel et al., 2006; Seplaki et al., 2006; ).”
2. In the Discussion section, lines 416-417, the Author mentioned that “… the number of cases used in the analysis is relatively small”. Please provide the reader with some numerical data in this context, please tell in how many cases (out of the analyzed cases) the intergenerational effect of socioeconomic status on depression symptoms was found.
The sentence “… the number of cases used in the analysis is relatively small” means that although this study used the large national sample (n=9,997) as the secondary data, the cases used in the analysis for this study was 378 cases which restricting the sample of pairs of G2 mothers and G3 daughters. I provided this information in the 2.1 Data and Sample section. I think it would be better to omit the sentence as it seems to confuse the reader.
Reviewer 3 Report
Dear Authors,
topics covered in your article are of current interest and sharing your experience can provide a contribution to a better understanding of intergenerational continuity of symptoms/diseases and to a more specific characterization of health inequality, thus providing adequate prevention and intervention strategies.
The article is well structured and is written in good English; the concepts are logically developed and exposed in a consistent sequential order. Methods and results are clearly described and well reported, and conclusions are adequately supported by the findings.
However, other variables should be included in the sample's description, if available, such as clinical information about general and mental health, more specifically the presence of diseases like mood disorders. Otherwise, their lack should be added to the Limitations section, together with “other socioeconomic variables” already reported by the authors.
I would suggest improving the background of the study and deepening description and discussion of depressive symptoms in the Introduction, approaching the topic in a more specific - psychiatric - way. In this view, authors might benefit from eventually referring at the following article: “Moccia L, Mazza M, Di Nicola M, Janiri L. The Experience of Pleasure: A Perspective Between Neuroscience and Psychoanalysis. Front Hum Neurosci. 2018 Sep 4;12:359. doi: 10.3389/fnhum.2018.00359”.
Further, I would suggest reporting the statistical significance of findings in the Abstract, in order to evidence the validity of the study and to give readers an overall and more complete first impression of the paper.
Finally, a revision of grammar is advisable to correct typos present in the manuscript, as well of style, to make some paragraphs more fluent by avoiding redundant information (e.g. 1.2 “The effects of childhood SES on depressive symptoms later”).
Best regards.
Author Response
Responses to Reviewer’s Comments
The Impact of Childhood and Adult Educational Attainment and Economic Status on Later Depressive Symptoms and It’s Intergenerational Effect (Manuscript ID: IJERPH-990995)
I appreciate the reviewer’s constructive feedbacks that held me clarify the important point.
1. Other variables should be included in the sample’s description, if available, such as clinical information about general and mental health, more specifically the presence of diseases like mood disorders. Otherwise, their lack should be added to the Limitations section, together with “other socioeconomic variables” already reported by the authors.
KLoWF did not provide the information about the clinical information about mental health. I described it as a study limitation in the discussion section.
2. I would suggest improving the background of the study and deepening description and discussion of depressive symptoms in the introduction, approaching the topic in a more specific – psychiatric – way. In this view, authors might benefit from eventually referring at the following article: “Mocca L, Mazza M, Di Nicola M, Janiri L. The Experience of Pleasure: A Perspective Between Neuroscience and Psychoanalysis. Front Hum Neurosci. 2018 Sep 4;12:359. Doi:10.3389/fnhum.2018.00359”.
I read the article the reviewer suggested. I think it is very useful to describe the mental health issue in South Korea. I put the sentence in background of this manuscript:
“Mental health problems prevalent in Korean women may deprive them of the pleasure of life and lower their motivation for entire life (Moccia, et al., 2018).”
3. Further, I would suggest reporting the statistical significance of findings in the Abstract, in order to evidence the validity of the study and to give readers an overall and more complete first impression of the paper.
I reported the statistical significance of findings in the Abstract.
4. Finally, a revision of grammar is advisable to correct typos present in the manuscript, as well of style, to make some paragraphs more fluent by avoiding redundant information (e.g., 1.2 “The effects of childhood SES on depressive symptoms later”)
I corrected typos in the manuscript and I removed the sentence which provided redundant information in 1.2 such as:
“Further, in the life-course approach, adult health is viewed as a consequence of differences in early life experiences (Adler et al., 2007; Luo & Waite, 2005; Reynolds & Ross, 1998). However,”